# Molecular Evidence for Flea-Borne Rickettsiosis in Febrile Patients from Madagascar

**DOI:** 10.3390/pathogens10111482

**Published:** 2021-11-14

**Authors:** Christian Keller, Raphaël Rakotozandrindrainy, Vera von Kalckreuth, Jean Noël Heriniaina, Norbert Georg Schwarz, Gi Deok Pak, Justin Im, Ligia Maria Cruz Espinoza, Ralf Matthias Hagen, Hagen Frickmann, Jean Philibert Rakotondrainiarivelo, Tsiry Razafindrabe, Denise Dekker, Jürgen May, Sven Poppert, Florian Marks

**Affiliations:** 1Institute of Virology, Philipps University, 35043 Marburg, Germany; 2Infectious Disease Department and Diagnostic Department, Bernhard Nocht Institute for Tropical Medicine, 20359 Hamburg, Germany; dekker@bnitm.de (D.D.); j.may@uke.de (J.M.); sven@poppert.eu (S.P.); 3Department of Microbiology and Parasitology, University of Antananarivo, Antananarivo BP 566, Madagascar; Rakrapha13@gmail.com (R.R.); herimandaniainatolotra@yahoo.fr (J.N.H.); rakotofifamanor@gmail.com (J.P.R.); rakouttsa@gmail.com (T.R.); fmarks@ivi.int (F.M.); 4International Vaccine Institute, 1 Gwanak-ro, Gwanak-gu, Seoul 08826, Korea; vera.vkalckreuth@web.de (V.v.K.); gdpak@ivi.int (G.D.P.); justin.im@ivi.int (J.I.); lcruz@ivi.int (L.M.C.E.); 5Independent Researcher, 67227 Frankenthal, Germany; Schwarznorbert@web.de; 6Department of Microbiology and Hospital Hygiene, Bundeswehr Central Hospital Koblenz, 56070 Koblenz, Germany; ralfmatthiashagen@bundeswehr.org; 7Department of Microbiology and Hospital Hygiene, Bundeswehr Hospital Hamburg, 20359 Hamburg, Germany; frickmann@bnitm.de; 8Institute for Medical Microbiology, Virology and Hygiene, University Medicine Rostock, 18057 Rostock, Germany; 9German Centre for Infection Research (DZIF), Hamburg-Lübeck-Borstel-Riems, 38124 Braunschweig, Germany; 10Department of Tropical Medicine II, University Medical Center Hamburg-Eppendorf (UKE), 20359 Hamburg, Germany; 11Cambridge Institute of Therapeutic Immunology and Infectious Disease, University of Cambridge School of Clinical Medicine, Cambridge Biomedical Campus, Cambridge CB2 0SP, UK

**Keywords:** *Rickettsia typhi*, *Rickettsia felis*, Madagascar, flea-borne rickettsiosis, murine typhus

## Abstract

Rickettsiae may cause febrile infections in humans in tropical and subtropical regions. From Madagascar, no molecular data on the role of rickettsioses in febrile patients are available. Blood samples from patients presenting with fever in the area of the capital Antananarivo were screened for the presence of rickettsial DNA. EDTA (ethylenediaminetetraacetic acid) blood from 1020 patients presenting with pyrexia > 38.5 °C was analyzed by *gltA*-specific qPCR. Positive samples were confirmed by *ompB*-specific qPCR. From confirmed samples, the *gltA* amplicons were sequenced and subjected to phylogenetic analysis. From five *gltA*-reactive samples, two were confirmed by *ompB*-specific qPCR. The *gltA* sequence in the sample taken from a 38-year-old female showed 100% homology with *R. typhi*. The other sample taken from a 1.5-year-old infant was 100% homologous to *R. felis*. Tick-borne rickettsiae were not identified. The overall rate of febrile patients with molecular evidence for a rickettsial infection from the Madagascan study site was 0.2% (2/1020 patients). Flea-borne rickettsiosis is a rare but neglected cause of infection in Madagascar. Accurate diagnosis may prompt adequate antimicrobial treatment.

## 1. Introduction

Rickettsiae are arthropod-borne, obligate intracellular Gram-negative bacteria. Phylogenetically, they are divided into four groups: the largely tick-associated spotted fever group that encompasses >25 species (e.g., *Rickettsia* [*R.*] *rickettsii*, *R. africae*), the lice-/flea-associated typhus group (e.g., *R. typhi*), the transitional group (e.g., *R. felis,* which is flea-associated) and the tick-associated ancestral group (e.g., *R. bellii*) [1,2]. 

Some, but not all rickettsiae harbored by arthropods are well-known human pathogens, and their pathogenicity differs from species to species. *R. rickettsii*, the agent of Rocky Mountain spotted fever (RMSF), is endemic in the Americas and is transmitted by ticks of the genera *Dermacentor* and *Rhipicephalus*. RMSF is probably the most lethal rickettsiosis, with a case-fatality rate of >35% when treatment with doxycycline is delayed [3]. In Brazil, mortality of RMSF may exceed 50% in some clusters [4]. On the African continent, entomological studies have confirmed the presence of rickettsial DNA in several different tick and flea species [5,6,7]. *R. africae*, which infests *Amblyomma* ticks to a percentage of >90%, is the causative agent of African tick-bite fever (ATBF). ATBF is often reported in travelers to Sub-Saharan Africa and usually manifests as a benign disease [8].

Fleas are typical vectors for *R. typhi* and *R. felis*. While *R. typhi* is mainly found in the rat flea *Xenopsylla* (*X.*) *cheopis*, *R. felis* has been detected in several flea species, including the cat flea *Ctenocephalides felis*, *X. cheopis* and the human flea *Pulex irritans* [6]. Molecular studies from Northern Africa have reported clinical cases of murine typhus [9,10]. Seroepidemiological reports from Sub-Saharan Africa suggest that up to 11% of the population are exposed to *R. typhi*, e.g., in Mali [11,12]. The mortality of murine typhus is very low (<1%) [13]. *R. felis* was identified as a potential cause of febrile illness in 1-15% of patients with fever from some African countries, e.g., Ghana and Senegal [14,15]. However, *R. felis* has also been found in blood samples of non-febrile control subjects, albeit at a significantly lower rate than in febrile patients [16,17]. Consequently, there is some controversy about the pathophysiology of *R. felis* infections and its role as an obligatory pathogen.

Clinically, spotted fever group rickettsioses often present with fever and an efflorescence called “eschar”, a cutaneous ulcerative lesion at the site of tick bite [18]. Typhus group rickettsioses such as murine typhus (*R. typhi*) and epidemic typhus (*R. prowazekii*) are not associated with eschar formation and are thus more difficult to diagnose. Murine typhus presents with a “classical” triad of fever, headache and rash, which is encountered in 33% of patients [19]. *R. felis* has been associated with similar symptoms as murine typhus [14,20]. 

Until recently, the importance of rickettsial infections in Madagascar has remained largely unknown. High percentages of *Amblyomma variegatum* (86.5%) and *Amblyomma chabaudii* ticks (100%) were found to be infested with *Rickettsia* (*R.*) *africae* [7,21]. *Xenopsylla cheopis* fleas, collected from small mammals in urban Malagasy areas, tested positive for *R. typhi* and *R. felis*, while *Pulex irritans* fleas from the same area were positive for *R. felis* [6]. Although it was shown that humans carry antibodies against both typhus group (TG) and spotted fever group (SFG) rickettsiae [6,21], direct molecular evidence for rickettsial infections in humans from Madagascar is so far lacking. A recent pan-Madagascan study that enrolled 682 febrile patients failed to detect rickettsial DNA by applying a novel macroarray system [22]. Here, we present the results of a real-time PCR-based screening for rickettsial DNA in EDTA blood of febrile patients from Madagascar.

## 2. Results

The patient population, a subset from the Typhoid Fever Surveillance in Africa Program (TSAP) study, was previously characterized in detail by Boone et al. [23]. Briefly, 1020 samples from patients living in a rather rural environment (Imerintsiatosika) or in urban slums (Isotry, central Antananarivo) and presenting with pyrexia ≥38.5 °C were selected for the present study. From 1,020 EDTA samples screened for rickettsial DNA by *gltA*-specific qPCR, five samples showed amplification in at least one of two replicate runs. Of those, 2/5 samples (#4133, #4196) were confirmed by repeated testing using the same target.

To identify the rickettsial species, *gltA* amplicons were sequenced applying Sanger sequencing. Excluding primer regions, a sequence of 116 bp was analyzed (for *gltA* sequences, see Table A1 and Table A2). The two positive samples differed in nine positions. Compared to published sequences in the NCBI database, the *gltA* amplicon of sample #4133 showed 100% homology to several reference strains of *R. typhi*, while sample #4196 showed 100% homology to *R. felis*. The phylogenetic analysis, which included other closely related *Rickettsia* species, confirmed this identification (Figure 1). 

The Ct values of >35 from the two blood samples suggested a low concentration of bacterial target DNA (Table 1). Thus, the molecular typing and confirmation approaches were different in both cases. The amplification of a larger partial *ompB* sequence (amplicon size: 855-861 bp) [24] was successful only for sample #4133 (in one sequencing direction). Sanger sequencing of the amplicon (GenBank OL310470) showed 100% homology with *R. typhi*, thereby confirming the *gltA* sequence result.

A larger *ompB* amplicon could not be obtained from sample #4196. Instead, we utilized the intergenic spacer-typing approach by Fournier et al. to confirm the *gltA* sequencing result [25]. The two intergenic spacers *dksA-xerC* (amplicon size: 164 bp, see Table A2) and *rpmE-tRNAfMet* (amplicon size: 351 bp, GenBank OL310471) were successfully sequenced from sample #4196. Both showed 100% homology with *R. felis*. The identification of the rickettsial species in sample #4196 as *R. felis* was further confirmed by a *R. felis*-specific *ompB* qPCR [26] (Table 1). 

Both patients with molecular confirmation of *Rickettsia* spp. in their blood samples were from the urban study site in Antananarivo. The *R. typhi* patient was a 38-year-old female from the Manarintsoa-Isotry district in central Antananarivo, while the *R. felis* patient was a 17-month-old infant from Antohomadinika Sud, another central area in Antananarivo. In both individuals, a fever of > 39 °C and a cough were noted, but no rash (Table 1).

These results provide a molecular confirmation of human-pathogenic rickettsiae for the first time in febrile patients from Madagascar. Only the flea-borne species *R. typhi* and *R. felis* were found; no tick-borne rickettsiae were identified. Overall, the positive rate for *Rickettsia* spp. DNA in febrile patients was 0.2% (2/1020 patients) in this study.

## 3. Discussion

The high endemicity of plague in Madagascar indicates that the Malagasy population is exceptionally exposed to flea-borne infections [27,28]. However, little is known about the regional epidemiology of flea-borne rickettsioses. Apart from a single case of an *R. typhi* infection in a French traveler returning from Madagascar [29], there has only been indirect serological evidence that humans may become infected [6]. Molecular data from fleas that showed infection with rickettsiae suggested that there might be a risk of transmission to humans [6]. The present study shows for the first time, based on a real-time PCR screening for rickettsial DNA in febrile patients, that human-pathogenic, flea-borne rickettsiae are transmitted to humans in Madagascar. 

Overall, the detection rate of rickettsial DNA in whole blood samples was low in this study (2/1020 patients). While we cannot exclude that storage and transport conditions may have affected the stability of target DNA in our study, previous investigations showed that *Rickettsia*-specific PCRs from blood samples may yield false-negative results in serologically confirmed cases [30,31]. A systematic review reported a blood DNA median sensitivity of 18% (6–27%) for pan-*Rickettsia* PCRs (SFG: 42% [6–69%]; TG: 3% [1–18%]) [32]. Thus, we may have missed some rickettsial infections by our approach. Importantly, the high Ct values identified here should not be interpreted as a sign of low clinical relevance, since they reflect the transient nature of these pathogens in circulating blood [33]. In sum, it is likely that the actual number of rickettsial infections among febrile patients may be several-fold higher than that detected by PCR.

With 1/1020 samples (0.1%) testing positive for *R. felis*, the detection rate in the urban setting of the Malagasy capital is lower than previously reported from continental African countries, where frequencies from 1.5% PCR-positive samples in Ghana to 15% in rural Senegal have been found [14,15]. As in our 1-year- and 5-month-old patients, *R. felis* is thought to be a common cause of febrile disease in children [14]. It can be detected significantly more often in febrile compared to afebrile patients, but its identification in afebrile pediatric control groups [16,34] has questioned its role as an obligatory pathogen. Another conundrum is the significance of *Plasmodium falciparum*/*R. felis* co-infections in regions with high malaria endemicity [14,15]. Notably, in our Malagasy study population, malaria was found only in a very low percentage of febrile patients (0–2%) [35]. 

*R. typhi* causes murine typhus, an infection with low mortality, and is transmitted worldwide, mainly in suburban areas [36]. Epidemiological studies indicate its transmission to humans in African countries [11,12,37], and direct molecular evidence from fever studies has primarily been provided from the northern African countries Algeria and Tunisia [38]. 

Infections with *R. typhi* and *R. felis* are associated with similar clinical presentations. Distinguishing them is possible by molecular diagnostics [26,39], but difficult by serology. In both infections, typical symptoms include fever and cough, as in our patients, and the absence of rashes often complicates the clinical diagnosis [14,36].

The present study found no evidence for spotted fever group rickettsiae in febrile patients. While according to seroprevalence studies, transmission of SFG rickettsiosis occurs in Madagascar [21], these infections are likely to only rarely present as bloodstream infections and may therefore be difficult to diagnose by applying molecular diagnostic approaches with EDTA blood samples.

While it was not determined in the present study whether fleas from the specific patients’ areas of residence carry *R. felis* or *R. typhi*, a similar study was performed in urban areas of the Tsiroanomandidy district, about 200 km west of Antananarivo. Here, 26/117 (24%) *Xenopsylla cheopis* fleas collected from small mammals tested positive for *R. typhi*, and 2/117 (2%) were positive for *R. felis*. In 8/26 (31%) *Pulex irritans* fleas, *R. felis* was detected. No co-infections with both species in the same flea were reported. It is highly likely that rodents in Antananarivo carry fleas with a similarly high proportion of *Rickettsia* infection, thus posing a risk for transmission of *R. felis* and *R. typhi* to humans. 

In future research, a prospective study design may be appropriate when investigating rickettsial infections in Madagascar. A comprehensive approach that takes into account clinical clues for rickettsial infections (e.g., eschars or rashes), serological assays (e.g., IgM/IgG against several rickettsial antigens) and immediate sample processing for PCR (e.g., from buffy coat) may provide a more complete picture of spotted and typhus fever rickettsioses.

## 4. Materials and Methods

### 4.1. Human Blood Sample Collection

EDTA blood samples were collected within the framework of the Typhoid Fever Surveillance in Africa Program (TSAP) [40]. A detailed description of how samples were collected is provided elsewhere [23]. In brief, 4500 EDTA blood specimens were collected from febrile patients in Madagascar between 2011 and 2013 continuously throughout the year. Of those, a total of 1,020 samples from individuals presenting with a body temperature ≥38.5°C were screened for the presence of rickettsial DNA. Samples were previously tested for the presence of *Burkholderia pseudomallei*, *Coxiella burnetii*, *Brucella* spp., *Leptospira* spp. and *Borrelia* spp. DNA, with a detection rate of 1.5% for *Brucella* spp. and otherwise negative results [23,41].

### 4.2. DNA Extraction and Real-Time PCR

Total DNA was extracted from 1 ml of human EDTA blood samples using the Flexigene DNA Kit (Qiagen, Hilden, Germany), following the manufacturer’s protocol. Rickettsial DNA was screened for by *gltA*-specific real-time PCR, using a modification of a previously published protocol based on the primers RKND03_F (5′-GTGAATGAAAGATTACACTATTTAT-3′) and RKND03_R (5′-GTATCTTAGCAATCATTCTAATAGC-3′), which yields an amplicon of 166 bp (base pairs) [42,43]. In a total of 10 µL, reactions contained 1x reaction buffer (Invitrogen, Darmstadt, Germany), 2.5 mM magnesium chloride, 0.2 mM deoxynucleotides (ABI), 5 nM bovine serum albumin, 0.5 µM of each oligonucleotide (Tibmolbiol, Berlin, Germany), SYBR Green, 0.25 U PlatinumTaq DNA polymerase and 2 µL of sample. Reactions were run in duplicate on a LightCycler 480 II instrument (Roche, Mannheim, Germany). The following cycling conditions were used: activation 95 °C, 15 min; 45 cycles: denaturation 94 °C, 15 s; annealing 60 °C, 30 s; elongation 72 °C, 30 s.

When a typical amplification curve was found, the result was confirmed in two independent, *ompB*-specific real-time PCRs for *R. typhi* and *R. felis* that allow detection of each species by a molecular beacon [26]. Samples showing amplification in both *gltA* and *ompB* assays were interpreted as positive for *Rickettsia* spp. 

### 4.3. Conventional PCR, Sequencing and Phylogenetic Analysis

For further typing, the intergenic spacer *dksA-xerC* was amplified using the primers dksA-xerC_F (5′-TCCCATAGGTAATTTAGGTGTTTC-3′) and dksA-xerC_R (5′-TACTACCGCATATCCAATTAAAAA-3′ [25]). The spacer *rpmE-tRNAfMet* was amplified using the primers rpmE-tRNArMet_F (5′-TTCCGGAAATGTAGTAAATCAATC-3′) and tRNArMet_F (5′-TCAGGTTATGAGCCTGACGA-3′ [25]). For sequencing a part of the *ompB* gene, the primers 120-M59 (5′-CCGCAGGGTTGGTAACTGC-3′) and 120-807 (5′-CCTTTTAGATTACCGCCTAA-3′ [24]) were used.

All conventional PCR reactions were run on a GeneAmp PCR System 2700 thermocycler (ABI) in a total volume of 50 µL, in the presence of 1x reaction buffer (Invitrogen, Darmstadt, Germany), 1.5 mM magnesium chloride, 0.2 mM deoxynucleotides (ABI), 0.2 µM of each oligonucleotide (Tibmolbiol, Berlin, Germany), 2 U PlatinumTaq DNA polymerase and 2 µL of sample. Upon denaturation at 95 °C for 3 min, 40 PCR cycles of 30 s at 95 °C, 30 s at 54 °C (spacers) or 58 °C (*ompB*) and 1 min at 72 °C were run, followed by a 5 min extension at 72 °C.

The *gltA*, *ompB*, *dksA-xerC* and *rpmE-tRNAfMet* amplicons were visualized by agarose gel electrophoresis and subjected to bidirectional conventional Sanger sequencing (Seqlab, Göttingen, Germany). Sequences <200 bp are provided in Appendix A (Table A1 and Table A2); sequences > 200 bp were deposited in the NCBI GenBank under the accession numbers OL310470 (*ompB*) and OL310471 (*rpmE-tRNAfMet*). 

Upon Clustal W multiple alignment (BioEdit Sequence Alignment Editor, version 7.0.5.3), *gltA* sequences were analyzed phylogenetically using the MEGA7 software (Molecular Evolutionary Genetics Analysis version 7.0 for bigger datasets; Kumar, Stecher, and Tamura 2015).

## 5. Conclusions

Taken together, the present study provides direct molecular evidence for both *R. typhi* and *R. felis* in blood samples of febrile patients from Madagascar. We suggest that flea-borne rickettsial infections are a relevant differential diagnosis of fevers of unknown etiology in a small number of patients and should be considered when deciding on empirical antimicrobial therapy by healthcare professionals in Madagascar.

## Figures and Tables

**Figure 1 pathogens-10-01482-f001:**
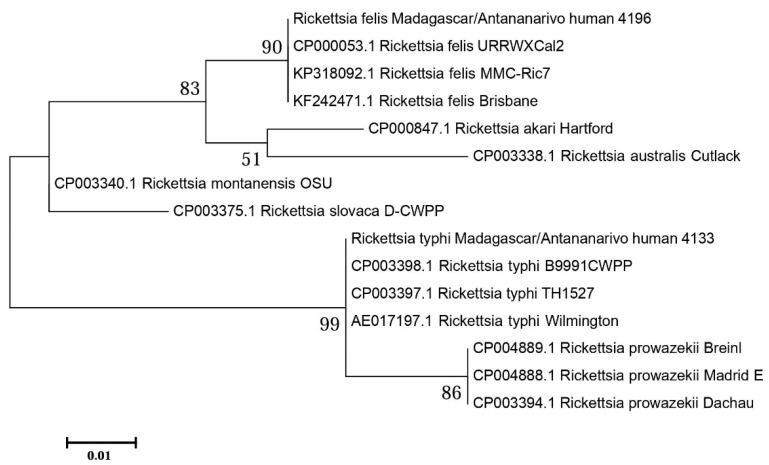
An amplicon of 116 bp from the rickettsial citrate synthase gene (*gltA*) was aligned with *gltA* sequences from rickettsial reference strains retrieved from the NCBI database, using Clustal W (BioEdit 7.0.5.3). A phylogenetic analysis was performed in MEGA7 using the maximum likelihood method (Tamura-Nei model). The percentage of trees in which the associated taxa clustered together is shown next to the branches.

**Table 1 pathogens-10-01482-t001:** Detection of *Rickettsia* DNA: sample and patient characteristics.

ID	Age	Sex	Residence	Body Temp.	Rash	Cough	*gltA* Ct	*gltA* Seq.	*R. felis ompB* Ct	*Rick.* *ompB* seq.
4133	38 y.	f	Manarintsoa Isotry (Antananarivo)	40.0	-	+	37.3	*R. typhi* (100%)	neg.	*R. typhi* (100%)
4196	1 y. 5 m	m	Antohomadinika Sud (Antananarivo)	39.4	-	+	35.2	*R. felis* (100%)	39.2	NP ^1^

^1^ NP = not possible. Ct = cycle threshold, y = year, m = month, ID = identification number, temp. = temperature, R. = Rickettsia, Rick. = Rickettsia spp.

## Data Availability

All relevant data are provided in the manuscript. The *ompB* sequence have been deposited at NCBI GenBank under accession number OL310470, the *rpmE-tRNAfMet* sequence is available under accession number OL310471.

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
