# Peer review of "Molecular Evidence for Flea-Borne Rickettsiosis in Febrile Patients from Madagascar"

_pathogens, 2021, doi:10.3390/pathogens10111482_

Round 1

Reviewer 1 Report

Overall

This study confirmed for the first time Rickettsia spp. detection in febrile patients in Madagascar. This is an important study and recommended for publication. There are several points that require the author's attention.

Major

Throughout manuscript: please correct the name and style of Rickettsia species as well as gene(s) mentioned in this manuscript.

Line 48-57: The introduction is too short. Suggest adding more information of the previous reports of rickettsiosis studies in the country including neighboring countries as well, although the author provided some information in the discussion part. 

L 76-87: Sequences of ompB as well as other genes (dksA-xerC and tRNAfMet) were mentioned but no sequences or data presented.

Fig 1: 

Figure legend is too long, please shorten it and only include important information. 

The program used is not consistent; in the legend it said MEGA6 but in the method was MEGA7.

More references should be included in the phylogenetic analysis, especially R. typhi sequences from French traveler (ref 11) and sequence from flea (ref 12).

GenBank accession number should be requested for the sequences generated from this study.

L133-135: the sentence is not clear. Why detection of rickettsial DNA in afebrile pediatric control groups has anything to do with the role of Rickettsia spp. And What does “an Obligatory pathogen” mean? Did the author mean “intracellular pathogen”? Please revise the sentence. 

L 144: R. felis is not a “Typhus group” even though it is quite similar to R. typhi to some extent. 

References:

Please check the reference style for this Journal, L. 256-257, as well as nomenclature for tick, pathogen, and others.. 

Minor

L 51-52: ...adding “respectively” at the end of the sentence.

Author Response

Reviewer 1

This study confirmed for the first time Rickettsia spp. detection in febrile patients in Madagascar. This is an important study and recommended for publication. There are several points that require the author's attention.

Comment 1:

Throughout manuscript: please correct the name and style of Rickettsia species as well as gene(s) mentioned in this manuscript.

Reply 1:

The name and styles of Rickettsia species and genes were italicized throughout the manuscript.

Comment 2:

Line 48-57: The introduction is too short. Suggest adding more information of the previous reports of rickettsiosis studies in the country including neighboring countries as well, although the author provided some information in the discussion part. 

Reply 2:

We agree with this reviewer that the introduction was too short in the previous version. We expanded this section; it now contains taxonomic, entomological and clinical aspects of rickettsial infections.

Comment 3:

L 76-87: Sequences of ompB as well as other genes (dksA-xerC and tRNAfMet) were mentioned but no sequences or data presented.

Reply:
In the previous version of the manuscript, sequence information was indeed lacking. We now provide all sequences in a supplementary file. We also submitted the sequences of ompB and the rpmE-tRNAfMet intergenic spacer to GenBank. The accession numbers are OL310470 and OL310471, respectively.

Comment 4:
Fig 1: Figure legend is too long, please shorten it and only include important information.
Reply:
The figure legend had been created as suggested by the authors of the MEGA7 software. However, we followed this reviewer’s suggestion to shorten the legend to the really essential information.

Comment 5:
The program used is not consistent; in the legend it said MEGA6 but in the method was MEGA7.
Reply 4:
We thank this reviewer for the identification of this inconsistency. The MEGA7 software was used for all analyses, which is now uniformly stated in both the methods and the legend.

Comment 5:
More references should be included in the phylogenetic analysis, especially R. typhi sequences from French traveler (ref 11) and sequence from flea (ref 12).
Reply 5:
This reviewer suggested adding more published sequences from Madagascar into the phylogenetic tree. While the suggestion is an excellent idea, both suggested publications did not provide gltA sequences that could have been included into the phylogenetic tree. Walter et al. detected rickettsial DNA by using a novel PCR targeting the Rpr 274P gene, while Rakotonanahary et al., published partial ompB sequences from fleas from Madagascar. We compared the four published R. typhi ompB sequences from fleas from Madagascar with the one obtained from our patient and found 100% identity on the nucleotide level (758 bp). However, since the ompB sequence of R. typhi is highly conserved – even across isolates from different continents – we decided not to include this information into the manuscript, since it would not add critical information.

Comment 6:
GenBank accession number should be requested for the sequences generated from this study.
Reply 6:
We agree with this reviewer that the sequences obtained by us should be made available via the GenBank data base. Respecting the lower limit of 200 bp at GenBank, we desposited the ompB and rpmE-tRNAfMet sequences in the GenBank database (accession numbers: OL310470 and OL310471). Additionally, we now provide supplementary data in the Appendix A containing the sequences <200 bp obtained in the present study.

Comment 7:
L133-135: the sentence is not clear. Why detection of rickettsial DNA in afebrile pediatric control groups has anything to do with the role of Rickettsia spp. And What does “an Obligatory pathogen” mean? Did the author mean “intracellular pathogen”? Please revise the sentence. 

Reply 7:
The point raised by this author is highly interesting. For obligatory pathogens, physicians generally assume that any given disease is directly caused by the present pathogen, e.g. via pathogenicity-associated genes. In the case of facultative pathogens, disease may be induced only in some patients. We cite two studies that have shown that R. felis can be detected in 3.3-3.4% of asymptomatic control groups. This finding naturally questions the role of R. felis as a pathogen. In the introduction, we now introduce the reader into this concept that is particularly important for understanding of R. felis.

Comment 8:
L 144: R. felis is not a “Typhus group” even though it is quite similar to R. typhi to some extent. 
Reply 8:
We agree with this reviewer that the previously used term was not strictly correct; we thus rephrased the passage.

Comment 9:
References: Please check the reference style for this Journal, L. 256-257, as well as nomenclature for tick, pathogen, and others.
Reply 9:
The citation of Sothmann et al. appeared to have no errors to us. The next reference (Mourembou et al.) however contained an error; we corrected it accordingly. We also checked the entire library for italicization of species names.

Comment 10:
L 51-52: ...adding “respectively” at the end of the sentence.
Reply 10:
We thank this reviewer for his / her suggestion. However, both Xenopsylla cheopis and Pulex irritans were found to be carriers of R. felis in fleas from Madagascar, so “respectively” does not fit here. To make the relationship clearer, we rephrased this sentence (now lines 86-87).

Reviewer 2 Report

Authors describe the detection of Rickettsia species for the first time in Madagascar using molecular means. Although the serological evidence for the same exists, species and phylogenetic classification of Rickettsia has been done for the first time in this study.

In the Methods, it is not mentioned when the PCR was put up. The samples were collected between 2011 and 2013. If the interval for the PCR was too much, it would have a bearing on overall positivity. The study design is retrospective.  If a prospective study is done, the detection might be higher.

What were the other infections for which the samples were tested?

What was the season when the patients for the original study were recruited?

Was the presence of eschar looked for? As samples were collected for Typhoid fever surveillance, this would not have been a parameter in the original study.

Have the authors deposited the PCR sequences in GenBank and obtained accession numbers?

In Discussion, line 126, how do the authors rule out a low clinical significance?

Author Response

Reviewer 2

Authors describe the detection of Rickettsia species for the first time in Madagascar using molecular means. Although the serological evidence for the same exists, species and phylogenetic classification of Rickettsia has been done for the first time in this study.

Comment 1:
In the Methods, it is not mentioned when the PCR was put up. The samples were collected between 2011 and 2013. If the interval for the PCR was too much, it would have a bearing on overall positivity. The study design is retrospective.  If a prospective study is done, the detection might be higher.
Reply:
The PCRs were performed in 2013-2015, between two and five years after the acquisition of the samples. In the meantime, the samples had been stored frozen at -20°C in Madagascar and later at -80°C in Germany. Although we generally agree that degradation of DNA may occur even under deep frozen conditions, experience by us and others indicate that the practical relevance for PCR diagnostics is low, in particular if short DNA sequences are targeted. Moreover, we have discussed the potential influence of storage and transport conditions in lines 119/120 (156-159 in the new version). As suggested by this reviewer, we now also discuss a prospective study design for future investigations (lines 198-203).

Comment 2:
What were the other infections for which the samples were tested?
Reply:
From the same EDTA blood samples, real-time PCR screening for B. pseudomallei, C. burnetii and Brucella spp. (Boone I, Henning K, Hilbert A, Neubauer H, von Kalckreuth V, Dekker DM, Schwarz NG, Pak GD, Krüger A, Hagen RM, Frickmann H, Heriniaina JN, Rakotozandrindrainy R, Rakotondrainiarivelo JP, Razafindrabe T, Hogan B, May J, Marks F, Poppert S, Al Dahouk S. Are brucellosis, Q fever and melioidosis potential causes of febrile illness in Madagascar? Acta Trop 2017; 172: 255-262) as well as for Leptospira spp. and Borrelia spp. (Hagen RM, Frickmann H, Ehlers J, Krüger A, Margos G, Hizo-Teufel C, Fingerle V, Rakotozandrindrainy R, Kalckreuth VV, Im J, Pak GD, Jeon HJ, Rakotondrainiarivelo JP, Heriniaina JN, Razafindrabe T, Konings F, May J, Hogan B, Ganzhorn J, Panzner U, Schwarz NG, Dekker D, Marks F, Poppert S. Presence of Borrelia spp. DNA in ticks, but absence of Borrelia spp. and of Leptospira spp. DNA in blood of fever patients in Madagascar. Acta Trop 2018; 177: 127-134.) had been performed in previous assessments. While 1.5% of the samples turned out to be positive for Brucella spp., no hints for the other targeted pathogens could be found. The information is now mentioned in lines 212-215.

Comment 3:
What was the season when the patients for the original study were recruited?
Reply:
The sampling was performed throughout the years of 2011 – 2013, as stated in the methods (line 210). No season was preferentially covered.

Comment 4:
Was the presence of eschar looked for? As samples were collected for Typhoid fever surveillance, this would not have been a parameter in the original study.
Reply:
We agree with this reviewer that the presence of an eschar can prompt the diagnosis of spotted fever rickettsiosis, even if rickettsial DNA cannot be detected from blood samples. Although a medical examination had been performed by a general practitioner during the inclusion of the patients into the study, there was no particular emphasis on thorough searches for eschar-like lesions including less obvious locations of the skin. Accordingly, the presence or absence of such lesions cannot be stated for sure. We included this point into the outlook for future research directions (lines 198-203).

Comment 5:
Have the authors deposited the PCR sequences in GenBank and obtained accession numbers?
Reply:
The NCBI GenBank requires that submitted sequences have at least 200 bp of length. With respect to this, we could not upload the gltA and dksA-xerC sequences which encompass only 116 bp and 117 bp, respectively. However, the longer sequences for ompB (759 bp) and rpmE-tRNAfMet (319 bp) were now deposited in the GenBank database. The accession numbers are OL310470 and OL310471, respectively.

Comment 6:
In Discussion, line 126, how do the authors rule out a low clinical significance?
Reply:
We thank this reviewer for giving us the opportunity to elucidate this point. We did not rule out a low clinical significance. What we stated in this paragraph is that a high Ct value by itself does not indicate a low significance, because of the transient nature of these pathogens in circulating blood. We now clarify this point by rephrasing this sentence (lines 161-163).

Reviewer 3 Report

The authors obtained blood samples from 1020 febrile patients and subjected them to gltA qPCR with Rickettsia specific primers to determine if some of these illnesses might be due to Rickettsia infection. The authors detected two cases for which exposure to flea-borne Rickettsia may be the cause.

1. Rickettsia felis is now considered to be in the transitional group of Rickettsia rather than the spotted fever group. I recommend papers from Joe Gillespie on that particular point, PMID: 19194535 for example. 

2. What is the point of this gltA fragment cladogram? Although this is the lone figure in the paper, it doesn't seem to contribute to the findings since simple blasting the Sanger sequencing result is enough to determine the closest species detected. It should probably be removed.

3. Several instances where R. felis / R. typhi should be in italics.

It would add to the paper if it was determined if these Rickettsia could be detected in fleas collected from the specific areas the positive patients were living. Given that R. felis has not previously been shown to cause illness but the authors I think correctly suppose that Rickettsia DNA might be transitory and difficult to detect in these samples, I also wonder if these fleas might simultaneously harbor R. typhi and R. felis

Author Response

Reviewer 3

The authors obtained blood samples from 1020 febrile patients and subjected them to gltA qPCR with Rickettsia specific primers to determine if some of these illnesses might be due to Rickettsia infection. The authors detected two cases for which exposure to flea-borne Rickettsia may be the cause.

Comment 1:
Rickettsia felis is now considered to be in the transitional group of Rickettsia rather than the spotted fever group. I recommend papers from Joe Gillespie on that particular point, PMID: 19194535 for example. 
Reply:
We now created additional introductory paragraphs. Within this paragraph, the grouping of R. felis is introduced to the reader. The suggested publication was cited, as suggested by this reviewer.

Comment 2:
What is the point of this gltA fragment cladogram? Although this is the lone figure in the paper, it doesn't seem to contribute to the findings since simple blasting the Sanger sequencing result is enough to determine the closest species detected. It should probably be removed.
Reply:
The percentage of sequence homology gives only partial information. The cladogram is a means to demonstrate that the obtained sequences do not only cluster with other R. felis / R. typhi but are also, upon unbiased ClustalW analysis, different from other closely related Rickettsia species. This distinction is possible despite the short length of the sequence. The cladogram was therefore kept in the manuscript and we respectfully ask the editor to accommodate this.

Comment 3:
Several instances where R. felis / R. typhi should be in italics.
Reply:
We thank this reviewer for careful reading of the manuscript. At all instances, species names were now italicized.

Comment 4:
It would add to the paper if it was determined if these Rickettsia could be detected in fleas collected from the specific areas the positive patients were living. Given that R. felis has not previously been shown to cause illness but the authors I think correctly suppose that Rickettsia DNA might be transitory and difficult to detect in these samples, I also wonder if these fleas might simultaneously harbor R. typhi and R. felis. 

Reply:
We agree with this reviewer that entomological studies from the patients’ residence areas would be highly intriguing. While we had no access to such a collection, a similar study was performed in an urban area 200 km west of Antananarivo (Rakotonanahary et al., 2017). The authors did not find co-infections, although the methodological approach enabled them to differentiate infections with either species. The results of this study and its implications for the present investigation are now discussed in the discussion part (lines 190-197).

Reviewer 4 Report

Authors detected Rickettsia felis and Rickettsia typhi from febrile patients and they provided first molecular evidence of human flea-borne rickettsiosis in Madagascar. They used blood samples from1,020 febrile patients, it’s quite large number. I felt that It’s really good work. However, manuscript need to improve especially introduction and materials and methods part.

Comments

Introduction part.

Introduction is too short. Authors need to put more information including general one for readers who does not have deep knowledge of Rickettsia. For example, explanation of Rickettsia organisms and their grouping in the 1st paragraph. Clinical symptoms of rickettsiosis, mortality of rickettsiosis, etc…in the 2nd t paragraph. 3rd paragraph should put explanation of the rickettsia distribution or epidemiological situation of rickettsiosis in Africa or Madagascar and near countries. Then, put the present paragraph. Like that.

Plz refer the introduction in this article for the 1st paragraph.

Thu, M.J., Qiu, Y., Matsuno, K. et al. Diversity of spotted fever group rickettsiae and their association with host ticks in Japan. Sci Rep 9, 1500 (2019). https://doi.org/10.1038/s41598-018-37836-5

Line 50.

“…infested with Rickettsia (R.) africae [1,2]…”

Remove “(R.)”

Line 51

“…for R. typhi and R. felis…”

Here, R. typhi and R. felis are appeared for the first time in the manuscript. Thus, do not omit.
 Should change to like that “…for Rickettsia typhi and Rickettsia felis…”

Results part

Line 64

“gltA” should be italic. Actually, through the manuscript there are many same problems.

In addition, here gltA is first appearance. Thus, here should be changed to

“…citrate synthase gene (gltA)-specific qPCR…”

Line 70, 73, 80, and 83,

“gltA” should be italic.

Line 74

“R. typhi” should be italic. Actually, through the manuscript there are many same problems.

Line 78

“ompB” should be italic. Actually, through the manuscript there are many same problems.

In addition, here ompB is first appearance. Thus, here should be changed to

“…outer membrane protein B gene (ompB)…”

Line 86.
“R. felis” should be italic. Through the manuscript there are many same problems.

Line 88, and 107.

“Rickettsia spp.” should change to “Rickettsia spp.” (“Rickettsia” should be italic.)

Discussion part

Author should discuss for the requirement of further study.

Line 132.

“17 months-old” in the sentence. But in table 1, author put “1year 5 month”. These must have the same notation.

Materials and Methods part

Authors mentioned “…two intergenic spacers dksA-xerC (amplicon size: 164 bp) and rpmE85 tRNAfMet (amplicon size: 351 bp) were successfully sequenced…” in the results part. However, there are no description for this method. How they get the sequence of these two ITS?  PCR or NGS or …? Author should describe the method for getting ITS sequences.

Furthermore, considering reproducibility of the research, authors should write more details of the methods, such as reaction condition (how many cycles? Annealing temperature?). Was ompB-qPCR same condition with gltA-qPCR?

Author Response

Reviewer 4

Authors detected Rickettsia felis and Rickettsia typhi from febrile patients and they provided first molecular evidence of human flea-borne rickettsiosis in Madagascar. They used blood samples from1,020 febrile patients, it’s quite large number. I felt that It’s really good work. However, manuscript need to improve especially introduction and materials and methods part.

Comment 1:
Introduction part. Introduction is too short. Authors need to put more information including general one for readers who does not have deep knowledge of Rickettsia. For example, explanation of Rickettsia organisms and their grouping in the 1st paragraph. Clinical symptoms of rickettsiosis, mortality of rickettsiosis, etc…in the 2nd t paragraph. 3rd paragraph should put explanation of the rickettsia distribution or epidemiological situation of rickettsiosis in Africa or Madagascar and near countries. Then, put the present paragraph. Like that.
Reply:
We thank this reviewer for his/her valuable suggestions to improve the introduction to our manuscript. As suggested, we added introductory paragraphs on Rickettsia organisms and their grouping as well as on the relationship between species, vectors and disease, as well as clinical symptoms and epidemiology.

Comment 2:

Plz refer the introduction in this article for the 1st paragraph. Thu, M.J., Qiu, Y., Matsuno, K. et al. Diversity of spotted fever group rickettsiae and their association with host ticks in Japan. Sci Rep 9, 1500 (2019). https://doi.org/10.1038/s41598-018-37836-5.
Reply:
We thank this reviewer for providing a literature source that explains the diversity of rickettsiae in their vectors. The study was added as a reference.

Comment 3:
Line 50. “…infested with Rickettsia (R.) africae [1,2]…” Remove “(R.)”
Reply:
In line 50 (now 51), the abbreviation “(R.)” is first used within the manuscript to abbreviate “Rickettsia”. According to international standards for scientific presentation, we kept this introduction.

Comment 4:
Line 51 “…for R. typhi and R. felis…” Here, R. typhi and R. felis are appeared for the first time in the manuscript. Thus, do not omit.  Should change to like that “…for Rickettsia typhi and Rickettsia felis…”
Reply:
We thank this reviewer for his/her suggestion. However, the introduction of species abbreviations should be performed once when the respective species is first introduced. This was the case line 51, when Rickettsia africae was mentioned (also see comment 3). Thus, we decided not to change this part.

Comment 5:
Results part, Line 64 “gltA” should be italic. Actually, through the manuscript there are many same problems. In addition, here gltA is first appearance. Thus, here should be changed to “…citrate synthase gene (gltA)-specific qPCR…”. Line 70, 73, 80, and 83, “gltA” should be italic.
Line 74. “R. typhi” should be italic. Actually, through the manuscript there are many same problems.
Line 78 “ompB” should be italic. Actually, through the manuscript there are many same problems. In addition, here ompB is first appearance. Thus, here should be changed to “…outer membrane protein B gene (ompB)…”
Line 86. “R. felis” should be italic. Through the manuscript there are many same problems.
Line 88, and 107. “Rickettsia spp.” should change to “Rickettsia spp.” (“Rickettsia” should be italic.)
Reply:
We thank this reviewer for giving us the opportunity to improve the writing. At all instances within the revised manuscript, gene names and species designations were now italicized.

Comment 6:
Discussion part. Author should discuss for the requirement of further study.
Reply:
We agree with this reviewer that we had not outlined the direction of future research. In the revised version of the manuscript, we now discuss potential future approaches how to study the role of rickettsioses in humans in Madagascar (lines 198-203).

Comment 7:
Line 132. “17 months-old” in the sentence. But in table 1, author put “1year 5 month”. These must have the same notation.
Reply:
In order to improve consistency throughout the manuscript, we followed this reviewer’s suggestion and now uniformly use “1 year 5 months”.

Comment 8:
Materials and Methods part. Authors mentioned “…two intergenic spacers dksA-xerC (amplicon size: 164 bp) and rpmE85 tRNAfMet (amplicon size: 351 bp) were successfully sequenced…” in the results part. However, there are no description for this method. How they get the sequence of these two ITS?  PCR or NGS or …? Author should describe the method for getting ITS sequences.
Reply:
We agree with this author that the manuscript had not revealed the methods regarding the ompB- and spacer-specific conventional PCRs. We therefore introduced a section 4.3. that explains how the spacer-specific PCRs and the ompB-specific PCR were run, including the respective primer sequences. The sequences have been submitted to GenBank; the accession numbers are OL310470 and OL310471, respectively.

Comment 9:
Furthermore, considering reproducibility of the research, authors should write more details of the methods, such as reaction condition (how many cycles? Annealing temperature?). Was ompB-qPCR same condition with gltA-qPCR?
Reply:
We thank this reviewer for giving us the opportunity to amend the required information. We now provide the requested information on real time and conventional PCRs in the method sections 4.2. and 4.3., with some corrections.

Round 2

Reviewer 1 Report

Excellent job on the added Introduction part. 

Only one comment to authors. Please check the number  of Table and Figure in the manuscript, since there is only one Figure and one Table, therefore, it should be Figure 1 and Table 1.

Reviewer 4 Report

Authors detected Rickettsia felis and Rickettsia typhi from febrile patients and they provided first molecular evidence of human flea-borne rickettsiosis in Madagascar. They used blood samples from1,020 febrile patients, it’s quite large number. I felt that It’s really good work.  

The revised manuscript was well written. I don't have any comments and suggestions, more.